# Sensitivity Analysis of Leakage Correction of GRACE Data in Southwest China Using A-Priori Model Simulations: Inter-Comparison of Spherical Harmonics, Mass Concentration and In Situ Observations

**DOI:** 10.3390/s19143149

**Published:** 2019-07-17

**Authors:** Zhiyong Huang, Jiu Jimmy Jiao, Xin Luo, Yun Pan, Chong Zhang

**Affiliations:** 1Department of Earth Sciences, The University of Hong Kong, Hong Kong 999077, China; 2The University of Hong Kong-Shenzhen Research Institute (SRI), Shenzhen 518057, China; 3The University of Hong Kong-Zhejiang Institute of Research and Innovation (HKU-ZIRI), Hangzhou 311305, China; 4Beijing Laboratory of Water Resources Security, Capital Normal University, Beijing 100048, China; 5State Key Laboratory Base of Urban Environmental Processes and Digital Modelling, Capital Normal University, Beijing 100048, China; 6State Key Laboratory of Earth Surface Processes and Resource Ecology, Faculty of Geographical Science, Beijing Normal University, Beijing 100875, China; 7Institute of Land Surface System and Sustainable Development, Faculty of Geographical Science, Beijing Normal University, Beijing 100875, China

**Keywords:** GRACE, leakage correction, forward modeling, a priori information, spherical harmonics, mascon, Southwest China

## Abstract

The Gravity Recovery and Climate Experiment (GRACE) level-2 spherical harmonic (SH) solutions are noisy and thus require filtering. Filtering reduces noise but affects signal quality via signal leakage. Generally, a leakage correction is required for GRACE applications to remove leakage signal and recover the true signal. Forward modelling based on some a priori information is a widely used approach for leakage correction of GRACE data. The a priori information generally relies on global hydrological model simulations. There are many global hydrological models and therefore it is of interest to explore how different global hydrology model simulations influence leakage correction results. This study investigated the sensitivity of three leakage correction methods (additive method, scaling factor method and multiplicative method) to five global hydrology model simulations (four models from the Global Land Data Assimilation System (GLDAS) and the WaterGAP Global Hydrology Model (WGHM)). The sensitivity analysis was performed with observational data in Southwest China and one sub-region, Guangxi. Results show that although large differences were identified among the five global model simulations, the additive and scaling factor methods are less affected by the choice of a priori model in comparison to the multiplicative approach. For the additive and scaling factor methods, WGHM outperforms the other four GLDAS models in leakage correction of GRACE data. GRACE data corrected with the multiplicative method shows the highest amount of error, indicating this method is not applicable for leakage correction in the study area. This study also assessed the level-3 mascon (mass concentration) solutions of GRACE data. The mascon-based results are nearly as good as the leakage corrected results based on SH solutions.

## 1. Introduction

The Gravity Recovery and Climate Experiment (GRACE) twin-satellite mission was a unique remote sensing technique that could monitor land surface mass changes by sensing the time-variable gravity field. GRACE satellite data have been widely used for various hydrological applications, such as disaggregating of the groundwater storage component from total terrestrial water storage anomaly (TWS) [1,2,3], estimating glacier and ice mass changes [4,5], and deriving total drainable water storage at basin scales [6,7]. Furthermore, GRACE data have been integrated into the depiction of hydrologic cycles, such as estimating evapotranspiration through terrestrial water balance [8,9] and estimating total basin discharge through combined land–atmosphere water balance computations [10]. In addition, GRACE data can be applied to drought and flood monitoring and characterization [11,12] and data assimilation for improving model parameterization [13,14].

Currently, the most widely used GRACE dataset is the level-2 spherical harmonic (SH) solutions, which are truncated at a certain degree and order (generally 96) with a typical footprint of 90,000 km^2^ [15,16]. Although substantial improvements (e.g., the update of background models) have been made on the GRACE data solutions [17], additional filtering is still necessary for removing the systematic stripes and high frequency random noises. Due to the truncation and filtering, noise can be reduced. Meanwhile, the true signals will be attenuated and will interrupt among the adjacent regions leading to signal leakage in a region of interest [18,19]. As a result, the apparent mass changes at the grid scale (1° × 1°) cannot represent the true signal. Instead, the signal at a grid is a mixture of true mass changes and leakage signals.

This study follows Longuevergne et al. [20] to define the terminology signal “leakage”, i.e., leakage for a certain region means the leakage-in signal from the exterior regions, while the attenuated true signal for a target region represents bias (or leakage-out). Leakage error is one of the many GRACE error components (e.g., measurement error and post processing error). In general, a leakage correction is required for GRACE applications to remove the impacts of leakage and recover the attenuated true signals (bias). The most widely used approach is forward modelling which uses the a priori information (generally from global model simulations or in situ observations) to mimic the GRACE data processing procedures, i.e., SH expansion and truncation, and filtering. The advantage of the forward modelling is that it can quantitatively assess the leakage errors of GRACE data. There are three different specifications on the forward modelling correction approach using the a priori information, i.e., the additive correction [8,12,21], multiplicative correction [22] and the scaling factor correction [23,24,25]. Also, there are some other methods that require no a priori information, such as the iterative forward modelling [26,27], the data driven approach [28] and the KeFin filter method [29]. Although level-3 GRACE data mascon (mass concentration) solutions [16,30,31,32] are now available, which require no leakage correction, they are not designed for a specific region. A recent study by Zhang et al. [33] demonstrated that mascon solutions can be used to recover mass redistribution of different mass sources in global and large basin scales (larger than approximately 3° × 3°), whereas the SH solution, when processed efficiently, can perform better than the mascon solution at smaller or local scales.

The interest of this study is the three forward modelling correction methods using the a priori information and the mascon solutions. The three leakage correction methods (additive correction, multiplicative correction and the scaling factor correction) have been widely applied in previous GRACE studies. For instance, the additive method has been used by Long et al. [12], Chen et al. [21] and Pan et al. [8] for correcting the TWS in the Yun-Gui Plateau of Southwest (SW) China, Upper Bramaputra River Basin and the Haihe River Basin, respectively. The leakage-corrected TWS from these previous studies is comparable to water balance estimates or in situ observations. Velicogna and Wahr [22] estimated the mass variations of the Antarctic ice sheet during 2002–2005 based on the multiplicative correction method. The scaling factor method was adopted by Famiglietti et al. [24] and Scanlon et al. [23] to correct the groundwater storage anomalies in the California Central Valley. Long et al. [25] used a state-of-the-art global hydrology model, PCR-GLOBWB, to derive scaling factors to restore the TWS for Yangtze River Basin, China. A study by Vishwakarma et al. [34] demonstrated that the scaling factor method was able to approach the ideal GRACE resolution with an acceptable error level of 2 cm.

Global model simulations of water storage changes (e.g., the models from Global Land Data Assimilation System (GLDAS)) have been widely used as the a priori information for leakage correction of GRACE data through forward modeling [8,12,21,35]. However, the model simulation itself has inevitable deficiencies, e.g., the incomplete representation of surface water, groundwater storage components and human impacts. Moreover, the simulations from different models can be largely different due to the differences of model structure, forcing data and parameters [36]. Therefore, the leakage correction of GRACE data might be sensitive to different a priori model simulations. However, limited studies (e.g., [37]) have evaluated the sensitivity of different leakage corrections to a priori model simulations, in particular based on in situ observations.

In this study we compare the performance of five global hydrology models and three leakage correction strategies, as well as the mascon solutions. Ground truth data of soil moisture and groundwater level in SW China are used to evaluate and validate GRACE estimates. SW China is a typical karst region with complex hydrogeology conditions which make it difficult to simulate the true TWS, in particular for groundwater storage variability using the GLDAS models and WGHM. Hence, there may be large uncertainty in the global model simulations. This provides the opportunity for answering the scientific question: how is the sensitivity of leakage correction of GRACE data (level-2 SH solutions) to various a priori model simulations? Also, this study aims to explore whether the leakage-corrected TWS based on the level-2 SH solution performs better than the level-3 mascon solution at the scale of SW China. In the following, the study region and data are presented in Section 2. Methods are illustrated in Section 3. Section 4 shows the results and discussion, and a brief summary is made in Section 5.

## 2. Study Region and Data

### 2.1. Study Region

The region of interest in this study is the SW China region (~802,900 km^2^), encompassing Guangxi (236,700 km^2^), Guizhou (176,200 km^2^) and Yunnan (390,000 km^2^) provinces (Figure 1). The latitude of SW China is between 21° N and 29° N. This is a largely populated region with a population of ~139.5 million. With a subtropical monsoon climate, SW China is humid and has a warm winter and hot summer. SW China receives abundant rainfall, which is unevenly distributed spatially and temporally. The annual rainfall is ~1200 mm/year with most rainfall between April and September. SW China is one of the largest karst areas globally with ~40% of karst landscapes (~316,000 km^2^). This is a region with complex mountainous topography and complicated hydrogeology conditions. The ecosystem of SW China is quite vulnerable, making it susceptible to flood and drought. Over the last decade, SW China experienced two extreme droughts, one from the autumn of 2009 to the spring of 2010, and another in the late summer of 2011. This study focuses on the large scale above 200,000 km^2^. In addition to the entire SW China region, this study selects one subregion, Guangxi province, for comparison, which is at a scale larger than 200,000 km^2^ and with dispersed in situ groundwater-level observations. Guizhou was not selected since it is smaller than Guangxi and below 200,000 km^2^, and Yunnan was not selected since the in situ groundwater-level observations are locally distributed in Kunming city (Figure 1).

### 2.2. GRACE Data

This study uses three level-2 release-5 (RL05) SH solutions and three level-3 mascon solutions (2003–2013) to calculate the TWS. The SH solutions are obtained from three scientific data centers, i.e., Center for Space Research (CSR) at the University of Texas, GeoForschungsZentrum (GFZ), Potsdam, and the Jet Propulsion Laboratory (JPL). For the SH products, the degree 0 and degree 1 stoke coefficients are removed, and the C20 terms are replaced using satellite laser ranging [38]. The P3M10 de-correlation method [39,40] is applied to remove the systematic longitudinal stripes, and a 200-km Gaussian smoother [18,41] is applied to reduce the random noise in stoke coefficients. The three mascon solutions are the CSR RL05 mascon [30], JPL RL05M mascon [16,31] and GSFC (Goddard Space Flight Center) mascon [32]. For SH solutions, a leakage correction is conducted following the method in Section 3.2. No filtering and leakage correction is required for the mascon solutions. A mean background gravity field from 2003 to 2013 is removed to retrieve the anomaly time series. There are several months with missing data (June 2003; January and June 2011; May and October 2012; and, March, August and September 2013), which are conventionally estimated through cubic-spline interpolation.

### 2.3. Global Model Simulations

This study uses the simulations from five global models, including four land surface models driven by the GLDAS-1, i.e., Mosaic, Noah, the Community Land Model (CLM) and the Variable Infiltration Capacity model (VIC) [42], and WGHM [43]. The four land surface models from GLDAS-1 simulate the soil moisture (SM) storage anomaly (SMS) with different layers at different depths. The CLM simulates SMS with 10 layers at the maximum depth of 3.433 m. The other three models, Noah, Mosaic and VIC, respectively simulate SMS at the depths of 2 m (4 layers), 3.5 m (3 layers), and 1.9 m (3 layers). All the outputs from these four GLDAS-1 models are at the resolution of 1° × 1°. The GLDAS-1 models do not simulate the surface water and groundwater storages.

WGHM is more sophisticated than the GLDAS-1 models, since it considers the components of surface water storage and groundwater storage, and computes groundwater recharge and river discharge (at 0.5 degree resolution) which are not incorporated in GLDAS-1 models [43]. One soil layer is modeled in WGHM with a soil–water balance considering the water content of the soil within the effective root zone, the effective precipitation, the actual evapotranspiration, and the runoff from the land surface [43]. A global lake and wetland database (GLWD) is used to simulate the surface water storage in WGHM [44]. Human impacts on water storage changes and river discharge are also simulated in WGHM [45]. In this study, the SMS simulations from GLDAS and the TWS (the summation of surface water storage anomaly, SMS and groundwater storage anomaly) simulations from WGHM are used as the a priori information for leakage correction of GRACE data. The contribution of snow water equivalent and canopy water storage are negligible.

### 2.4. In Situ Observations

The in situ observation data used in this study include the in situ SM, groundwater-level and reservoir water storage measurements from Huang et al. [46]. There are 18 in situ SM observations (6 in Yunnan, 5 in Guizhou and 7 in Guangxi, Figure 1) which are obtained from the China Meteorological Data Website. The SM data are in terms of the degree of relative soil saturation (i.e., the volumetric SM content divided by the soil porosity). The details on the estimation of observed SMS from relative soil saturation are shown in the supporting information of Huang et al. [46].

There are 37 in situ groundwater-level observations (15 in Yunnan, 13 in Guizhou and 9 in Guangxi, Figure 1) which are obtained from the groundwater-level Yearbook compiled and published by the China Institute of Geo-Environment Monitoring (CIGEM) [47]. Among the 37 wells, 30 are located at the carbonate karstic aquifers, 3 at the consolidated fractured aquifers, and 4 at the unconsolidated porous aquifers. The in situ groundwater storage anomalies are estimated by multiplying groundwater-level anomalies with the specific yields of the unconfined aquifers or the storage coefficients of the confined aquifers [48]. Details on the determination of specific yields and storage coefficients can be found in Section 2.4 of Huang et al. [46].

Due to the lack of in situ surface water storage data in lakes and wetlands, this study uses the simulation data from WGHM as a surrogate. Since WGHM does not simulate the water storage changes in the largest reservoir in SW China, the Longtan Reservoir, this study uses the water-level data from previous studies in this reservoir [49,50]. The reservoir water level data are converted to water storage according to the fitted exponential relation between observed water-level data and reservoir storage data provided by the Water Regime Annual Report of the Pearl River [51]. Details can be found in Section 2.3 of Huang et al. [46]. The in situ observations are used to estimate TWS. Then, the in situ TWS is estimated by summing up in situ SMS, groundwater storage, Longtan Reservoir storage and WGHM-simulated surface water storage.

This study assumes 40% uncertainty in the specific yield and storage coefficient [52] for estimating in situ groundwater storage anomalies (SW China: 17.3 mm/mo., Guangxi: 28.6 mm/mo.) and assumes 30% uncertainty for in situ SMS (SW China: 4.7 mm/mo., Guangxi: 6.9 mm/mo.) and WGHM-simulated surface water storage (SW China: 2.8 mm/mo., Guangxi: 3.5 mm/mo.). The uncertainty for the Longtan Reservoir storage induced by curve fitting is estimated to be ∼5% (SW China: 0.3 mm/mo., Guangxi: 1.1 mm/mo.). The final error for estimated in situ TWS is the propagated error (SW China: 18.2 mm/mo., Guangxi: 29.7 mm/mo.) of different storage components, i.e., calculating the root square of sum of the square errors.

## 3. Methods

### 3.1. Estimation of Regional-Averaged GRACE TWS

For the SH solutions, an exact regional kernel function [20,53] which is truncated at degree 60 and filtered using a 200-km Gaussian smoother at 0.5° resolution is applied to obtain the regional averaged TWS:(1)$$v\;\left(X\right)=\;\{\begin{array}{c} 1,\;\;\;\;\;\mathrm{if}\;X\in R\;\;\;\;\;\;\;\;\;\;\\ 0,\;\;\;\;\mathrm{if}\;X\in \omega -R\;\; \end{array}$$
where *X* is the position on the earth’s surface X=(θ, λ) and ω represents the entire earth surface, and *R* is the region of interest with an area of *S_region_*. The filtered exact kernel functions for SW China and its subregion Guangxi are shown in the Appendix A.

According to time-variable gravimetric theory [18,20], the regional-averaged TWS can be estimated using the following equation:(2)$${\hat{\bar{S}}}_{0}=\frac{a\rho_{e}}{3\Omega_{\mathrm{region}}\rho_{w}}\overset{60}{\underset{l=0}{\sum }}\overset{l}{\underset{m=0}{\sum }}\frac{2l+1}{1+k_{l}}W_{l}\left(v_{lm}^{c}\Delta C_{lm}+\;v_{lm}^{s}\Delta S_{lm}\right)$$
where ${\hat{\bar{S}}}_{0}$ is the regional-averaged TWS in terms of equivalent water height; a and $\rho_{e}$ are the mean radius and density (5517 kg/m^3^) of the earth; $\rho_{w}$ is the mean density of water (1000 kg/m^3^); $\Omega_{\mathrm{region}}$ is the singular area of the region ($\Omega_{region}=\;S_{region}/a^{2}$) and $S_{\mathrm{region}}$ is the area of the region of interest; $W_{l}$ is the smoothing coefficient of the Gaussian filter at degree *l*; $∆C_{lm}$ and $∆S_{lm}$ are the stoke coefficients from GRACE; $k_{l}$ is the load potential Love number at degree *l*; and $v_{lm}^{c}$ and $v_{lm}^{s}$ are the SH coefficients describing the exact kernel function:(3)$$v\left(\theta ,\lambda \right)=\frac{1}{4\pi }\overset{60}{\underset{l=0}{\sum }}\overset{l}{\underset{m=0}{\sum }}{\bar{P}}_{lm}\left(\cos\theta \right)\left(v_{lm}^{c}\Delta C_{lm}+\;v_{lm}^{s}\Delta S_{lm}\right)$$
(4)$$\left\{\begin{array}{c} v_{lm}^{c}\\ v_{lm}^{s} \end{array}\right\}=\int^{\;}v\left(\theta ,\lambda \right)\;{\bar{P}}_{lm}\left(\cos\theta \right)\;\left\{\begin{array}{c} \cos\;m\lambda \\ \sin\;m\lambda \end{array}\right\}\;ds$$

The estimation of TWS for a region of interest using Equation (2) can also be expressed as:(5)$${\hat{\bar{S}}}_{0}=\frac{1}{S_{region}}\int_{\omega }^{\;}\hat{\bar{S}}\;v\;d\omega$$
where $\hat{\bar{S}}$ is the filtered global GRACE TWS.

### 3.2. Leakage Correction Methods

The filtering process of GRACE data not only reduces high frequency noise, but also causes signal bias (i.e., attenuation of the true TWS signals inside the target region) and leakage (i.e., leakage-in signal from exterior region). Therefore, the filtered regional GRACE TWS can be expressed as follows [20,37]:(6)$${\hat{\bar{S}}}_{0}=\;{\bar{S}}_{0}-S_{B}+S_{L}$$
where ${\bar{S}}_{0}$ is the true signal, $S_{B}$ is the bias and $S_{L}$ is the leakage-in. In this study, three leakage correction methods are investigated to retrieve the true TWS time series $\bar{S_{0}}$. The studies by Long et al. [37] and Longuevergne et al. [20] have made detailed descriptions of these leakage correction methods, which are briefly summarized as follows.

The first method is the additive method [8,20,37,54]. This method requires the bias and leakage time series to be estimated separately through forward modeling using the global model simulations as the a priori information. The leakage-in signals are removed and the bias signals are added. The true signal can be estimated using the following Equation (7):(7)$${\bar{S}}_{0}={\hat{\bar{S}}}_{0}+S_{B}-S_{L}$$

The second method is the multiplicative correction [22]. This method first requires the leakage-in signal to be removed through forward modeling using the global model simulations as the a priori information. Then an assumption of a uniform mass distribution inside the region of interest is made to estimate a multiplicative factor *k*_m_:(8)$$k_{m}=\frac{1}{S_{region}\int_{s}^{\;}v\;d\omega }\;$$

The leakage-corrected GRACE TWS using the multiplicative correction can be estimated as:(9)$${\bar{S}}_{0}=\left({\hat{\bar{S}}}_{0}-S_{L}\right)\cdot \;k_{m}\;$$

Using Equation (8), the multiplicative factor is estimated to be 1.59 and 2.2 for SW China and Guangxi, respectively.

The third method for leakage correction of GRACE data uses the scaling factor [23,24,25]. For this method, the a priori global model simulations of TWS are processed in the same way as GRACE data, i.e., performing a SH expansion and truncating at degree and order 60, filtered with a 200 km Gaussian radius. Then a scaling factor *k* is calculated as the least-squares fitted coefficients between the filtered (or forward-modelled) and unfiltered water storage anomaly series:(10)$$M=\;\overset{T}{\underset{i=1}{\sum }}{\left({\bar{S}}_{i}-k{\hat{\bar{S}}}_{i}\right)}^{2}$$
where *M* represents the objective function to be minimized; ${\bar{S}}_{i}$ and ${\hat{\bar{S}}}_{i}$, respectively, are the unfiltered and filtered TWS through forward modelling for month *I*; and *T* is the total number of months from January 2003 to December 2012.

Based on the scaling factor *k*, the true TWS is obtained as follows:(11)$${\bar{S}}_{0}=k{\hat{\bar{S}}}_{0}$$

## 4. Results and Discussion

### 4.1. Modelled SMS versus In Situ Observations

Figure 2 shows the monthly time series of SMS anomalies from four GLDAS land surface models and the WGHM with comparison to in situ observations. The SMS anomalies from CLM and WGHM exhibit closer agreement to observed SMS than Noah, Mosaic and VIC. For the entire SW China region, the comparison between the modeled SMS anomalies from CLM (WGHM) and observed SMS anomalies shows a satisfactory correlation coefficient (r) of 0.63 (0.46) and acceptable root mean square error (RMSE) of 13.5 (17.6) mm (Figure 2a and Table 1). A similar comparison can also be found in Guangxi with r = 0.64, RMSE = 18.5 mm for CLM, and r = 0.61, RMSE = 19.4 mm for WGHM (Figure 2c and Table 1). However, the modeled SMS anomalies from Mosaic, Noah and VIC show much higher amplitudes (Figure 2b,d) than the observed SMS anomalies with significantly larger RMSE (>40 mm) than CLM and WGHM (Table 1). Even a high RMSE of 108.2 mm was found for Mosaic-modeled SMS anomalies in Guangxi compared to observed SMS anomalies (Table 1). Moreover, the SMS simulated by CLM and WGHM agree with each other very well with r = 0.83, RMSE = 10.1 mm for SW China, and r = 0.86, RMSE = 11.2 mm for Guangxi.

Figure 3 further compares the modeled SMS to observed TWS. For SW China, the SMS simulated by Mosaic, Noah and VIC shows better agreement (higher r and lower RMSE, see Table 1) with observed TWS than with observed SMS. For Guangxi, the SMS simulated by Mosaic shows better agreement (higher r and lower RMSE, see Table 1) with observed TWS than with observed SMS. Among the four GLDAS models, Noah-modeled SMS represents the best agreement (r = 0.87, RMSE = 30.8 mm) against observed TWS for the entire SW China region, followed by VIC, CLM and Mosaic. For Guangxi, a considerably larger discrepancy on the amplitude can be noted between modeled SMS and observed TWS with RMSE > 75 mm. Nonetheless, Noah-modelled SMS shows relatively better agreement with observed TWS than the other three GLDAS models. Better agreement is obtained between WGHM-modeled TWS and observed TWS for the entire SW China region (r = 0.86, RMSE = 34.8 mm) than that for Guangxi (r = 0.64, RMSE = 79.1 mm) (Table 1). The SMS from the four GLDAS models and the TWS from WGHM are used as the a priori information for leakage correction of GRACE data. The five a priori model simulations show an apparent difference with a standard deviation of 27.6 mm for SW China and 43.2 mm for Guangxi.

It is notable that modeled SMS (for Mosaic, Noah and VIC) is more in line with observed TWS rather than observed SMS. Similar comparison can also be found in previous studies (e.g., [55]) in which GLDAS model simulations were used to validate GRACE TWS. In the study of Yang et al. [55], the GLDAS SMS is used to represent TWS when comparing with GRACE TWS. It is reasonable to note a large amplitude of GLDAS SMS comparable to GRACE TWS considering the structure of GLDAS models. GLDAS models do not include a groundwater module, and they cannot output the information of the groundwater component into an independent layer. Instead, the information of the groundwater component is lumped into the soil moisture layer through a simple “bucket” model [56]. As a result, the SMS simulations contain the information of the groundwater component. Hence, the SMS is comparable to TWS.

### 4.2. Uncorrected GRACE TWS versus In Situ Observations

Figure 4 compares the uncorrected TWS from three GRACE SH products with the observed TWS. For SW China, the TWS from CSR shows the best agreement with observed TWS with the highest r of 0.93 and the lowest RMSE of 27.0 mm, compared to JPL (r = 0.93, RMSE = 29.7 mm) and GFZ (r = 0.86, RMSE = 37.2 mm) (Table 1). For Guangxi, the TWS from JPL shows the best agreement with observed TWS with the highest r of 0.80 and lowest RMSE of 61.7 mm, compared to CSR (r = 0.79, RMSE = 63.1 mm) and GFZ (r = 0.71, RMSE = 71.1 mm) (Table 1). From Table 1 it can be seen that GRACE performs better in monitoring TWS changes than GLDAS models and WGHM, with better agreement with in situ observations. As can be seen, even without leakage correction, appreciable agreement is found between GRACE TWS and in situ observations. Therefore, it is of interest to explore how the leakage correction approaches would further improve the TWS signals and to understand the uncertainties incurred by leakage correction. The comparison among the three GRACE estimates of TWS shows r ranging from 0.88 to 0.97 and RMSE below 36 mm, indicating that the overall difference among the three estimates is small. In the following analysis, the result from CSR is used since its uncertainty is smaller than JPL and GFZ [2,57].

### 4.3. Scaling Factors Derived from Different Models

Figure 5 shows the gridded scaling factor derived from four GLDAS models and WGHM. The scaling factors from different models show apparent differences and spatial heterogeneity. The gridded scaling factors from Noah show the lowest spatial heterogeneity with the coefficient of variance (CV) of 0.14, followed by CLM (CV: 0.18). The gridded scaling factors from Mosaic show the highest spatial heterogeneity with CV of 0.34. The spatial heterogeneity for the gridded scaling factors from VIC and WGHM is also relatively high, with CV of 0.28 and 0.30, respectively (Figure 5d,e). The gridded scaling factors from Noah range from 0.76 to 1.52, with the value of most of the grids larger than 1, indicating that the forward modeled signal of Noah in SW China is dominated by bias (leakage-out) signal, which needs to be corrected using scaling factors greater than 1. The gridded scaling factors from CLM range from 0.51 to 1.26, with most of the grid values within 0.6 and 1 (Figure 5a), indicating that the forward modeled signal of CLM is dominated by leakage-in signal, which needs to be corrected using scaling factors smaller than 1. With the highest spatial heterogeneity, the scaling factors from Mosaic range from 0.32 to 2.02, with the value in Guizhou and Guangxi larger than 1, while the value in Yunnan is smaller than 1 (Figure 5b), indicating that forward modeling using Mosaic simulations as the a priori information will lead to different dominant signals (leakage-in or leakage-out) in different regions. The scaling factors from VIC range from 0.25 to 1.82, with most of the grid values larger than 1. The scaling factors from WGHM range from 0.32 to 1.67, with most grid values smaller than 1, in particular for northern Guizhou where the scaling factors are below 0.6 (Figure 5e). The spatial distribution of scaling factors is different among different models. From Figure 5, it is apparent that the large values of Mosaic/VIC scaling factors are randomly distributed, while WGHM’s similar signals are spatially localized. This is a reflection of the different spatial distribution of a priori model simulations, which is caused by the different forcing data, parameterization and structure of the models [36].

Considering the large footprint of GRACE satellites, rescaling of filtered GRACE TWS at grid scale (1° × 1°) may be problematic [25]. In this study, two types of scaling factors, namely the regionally integrated and the spatially averaged scaling factors, are compared. For the regionally integrated approach, the scaling factor for a region of interest is derived by the least square fit between spatially averaged filtered and unfiltered modeled TWS time series. For the spatially averaged approach, the gridded scaling factors at 1° × 1° scale are firstly derived (as illustrated above). Then the spatially averaged scaling factor can be used as a proxy of the scaling factor for a region. Table 2 compares the regionally integrated and the spatially averaged scaling factors for SW China and Guangxi. For SW China, the regionally integrated and spatially averaged scaling factors from CLM are the lowest and smaller than 1 (0.72 and 0.88 respectively) among the five models, showing the largest relative difference (22%) (Table 2). The scaling factors from WGHM are also smaller than 1 (regionally integrated: 0.79, spatially averaged: 0.92) with the second largest relative difference (16%). The scaling factors from Mosaic, Noah and VIC are close to 1 with the lowest for Noah (0.97) and largest for VIC (1.14). The relative difference between the regionally-integrated and spatially averaged scaling factors for Mosaic, Noah and VIC are 9%, 12% and 15%, respectively (Table 2). For Guangxi, the scaling factors from CLM are the lowest (regionally-integrated: 0.97, spatially averaged: 0.99) with the lowest relative difference (only 2%). The scaling factors from Mosaic are the largest (regionally-integrated: 1.34, spatially averaged: 1.21) with the highest relative difference (10%) (Table 2). It can be noted that the scaling factors for Guangxi are slightly larger than those for the entire SW China region. This is normal considering the smaller spatial scale of Guangxi, which can result in more leakage-out signals and hence requires larger scaling factors to recover the biased signals. Overall, most of the scaling factors from the five models are close to 1. This indicates that the signal leakage effects in SW China and Guangxi are small and the derivation of scaling factors is insensitive to the a priori model simulations. Considering the relatively large spatial heterogeneity of the gridded (spatially averaged) scaling factor as illustrated above, this study only uses the regionally-integrated scaling factors to correct the GRACE TWS.

### 4.4. Leakage-Corrected and Mascon-Based GRACE TWS versus In Situ Observations

Figure 6 and Figure 7 display the residuals between uncorrected/leakage-corrected GRACE TWS and in situ observations of TWS for SW China and Guangxi, respectively. The original TWS time series before and after leakage correction compared to in situ observations are shown in the Supplementary Material (see Appendix A). Relevant statistics on r, RMSE and Nash–Sutcliffe efficiency (NSE) are shown in Table 3. For the entire SW China region, the results derived from additive correction using CLM (r = 0.93, RMS E= 25.0 mm, NSE = 0.84) and WGHM (r = 0.93, RMSE = 24.3 mm, NSE = 0.85) simulations as the a priori information show some improvement, with the RMSE reduced by 7% and 10%, respectively. For the scaling factor correction in SW China, the results derived using the CLM (r = 0.93, RMSE = 24.8 mm, NSE = 0.84) and WGHM (r = 0.93, RMSE = 23.8 mm, NSE = 0.86) simulations as the a priori information show some improvement with the RMSE reduced by 8% and 12%, respectively (Table 3). For the additive and scaling factor corrections in Guangxi, no apparent changes to r, RMSE and NSE can be noted, indicating no improvement to the results after leakage correction. As can be seen, the additive and scaling factor corrections perform better in the entire SW China region than in Guangxi, with larger improvement in the RMSE. This is owing to the larger spatial scale of SW China, which results in smaller leakage error, and which is hence more easily corrected. However, the multiplicative correction makes the results worse compared to in situ observations rather than improving the results. For SW China, the RMSE increases from 27.0 mm before correction to the range of 32.9–44.9 mm after multiplicative correction, and the r decreases from 0.93 before correction to the range of 0.84–0.92 after multiplicative correction (Table 3). For Guangxi, the RMSE increases from 63.1 mm before correction to the range of 73.4–100.3 mm after multiplicative correction, and the r decreases from 0.79 before correction to the range of 0.54–0.78 after multiplicative correction (Table 3). Therefore, the multiplicative correction method incurs errors and is not applicable for leakage correction of GRACE data over SW China and Guangxi. This may be owing to the fact that the multiplicative method assumes uniform distribution of mass changes inside of a region, and ignores the spatial heterogeneity of the actual mass distribution.

As illustrated above, the results derived using WGHM as the a priori information agree the best with in situ observations. This may be owing to the fact that WGHM simulates groundwater storage variations while other GLDAS models do not. A statistic is further made using the standard deviation of the five leakage-corrected TWS time series derived using the different model simulations as the a priori information. The standard deviation represents a conservative estimation of the leakage correction error for GRACE data. For SW China, the standard deviation for the additive, scaling factor and multiplicative methods is 9.0 mm, 9.6 mm, 15.8 mm, respectively; for Guangxi, the standard deviation is 14.8 mm, 11.7 mm, 33.4 mm, respectively. As can be seen, the additive and scaling factor methods show less leakage correction errors than the multiplicative method. The standard deviation for Guangxi is larger than that for the entire SW China region, indicating increasing leakage correction error at a smaller scale. Overall, although there is large uncertainty among the model simulations (Section 4.1), there is appreciable consistency (smaller standard deviation) among the results derived from the additive and scaling factor methods, which indicates that the two methods are less sensitive to the a priori information. However, the multiplicative method has relatively higher sensitivity to the a priori information.

Figure 8 compares the mascon-based TWS with in situ observations. As demonstrated earlier, the multiplicative correction method incurs errors and is not applicable for leakage correction of GRACE data over SW China and Guangxi. SH-based TWS with multiplicative correction shows poorer results than the mascon-based TWS with lower r and NSE, and higher RMSE (see Table 3). Overall, mascon-based TWS and SH-based TWS with additive and scaling factor correction show very similar results. The TWS from CSR mascon, JPL mascon and GSFC mascon, respectively, shows a correlation of 0.92, 0.92 and 0.89, a RMSE of 25.8 mm, 31.5 mm and 32.9 mm, and a NSE of 0.83, 0.75 and 0.73 (Table 3) compared to in situ observations for the entire SW China region. The correlation is within the range of 0.89–0.93, RMSE is within the range of 23.8–34.2 mm and NSE is within the range of 0.70–0.86 (Table 3) for the comparison between TWS with additive and scaling factor correction and in situ observations. The ranges of r, RMSE and NSE are similar between mascon-based TWS and TWS with additive and scaling factor correction when compared to in situ observations. Therefore, there is no apparent improvement for the mascon-based TWS relative to the SH-based TWS over the entire SW China region. For Guangxi, no improvement can be found for the CSR and GSFC mascons. However, TWS from the JPL mascon shows a correlation of 0.82 and RMSE of 58.3 mm (Table 3), which is better compared to in situ observations than any of the leakage-corrected TWS using the SH solutions, indicating slight improvement. The improvement found can be attributed to the strict spatial constraints applied in the JPL mascon solution [16,31].

## 5. Conclusions

This study assessed the TWS in SW China using spherical harmonics (SH) and mascon solutions, and investigates the sensitivity of leakage correction of GRACE data (level-2 SH) to the a priori information. Three leakage correction methods (the additive, scaling factor and multiplicative methods) under the a priori information of five model simulations, namely four GLDAS models (CLM, Mosaic, Noah and VIC) and WGHM, are systematically examined in the SW China region and one sub-region, Guangxi. Firstly, the accuracy of the a priori model simulations is evaluated using in situ observations. Apparent differences are found among the SMS simulations from the five models. The SMS from CLM and WGHM agree with each other well and are comparable to in situ SMS. However, the SMS from Mosaic, Noah and VIC overestimates the amplitude compared to in situ SMS. Modeled SMS (for Mosaic, Noah and VIC) is more comparable to observed TWS rather than observed SMS. Secondly, GRACE TWS (level-2 SH solution) without leakage correction is evaluated using in situ observations and model simulations. Results indicate that GRACE performs better in monitoring TWS changes than GLDAS and WaterGAP models. Thirdly, the scaling factors are evaluated at grid, regionally integrated and spatially averaged scales. Overall, most of the scaling factors from the five models are close to 1, indicating that the signal leakage effects in SW China and Guangxi are small and the derivation of scaling factors are insensitive to the a priori model simulations.

Results with leakage correction show that different leakage correction methods have different sensitivity to the a priori model simulations. Although there are large differences among the five a priori model simulations, the additive and scaling factor methods are less sensitive to the a priori model simulations. For these two methods, the standard deviation of the leakage corrected TWS using the five model simulations as the a priori information is less than 15 mm, which is smaller than the standard deviation (27.6 mm for SW China and 43.2 mm for Guangxi) of the five priori model simulations. For additive and scaling factor methods in the entire SW China region, the result derived using the WGHM simulations as the a priori information shows the best performance, with the RMSE (relative to in situ TWS) reduced by 10% and 12%, respectively. However, no apparent improvement can be noted for the results derived using the other four GLDAS models as the a priori information. This is because WGHM simulates groundwater storage variations while other GLDAS models do not. The additive and scaling factor corrections perform better in the entire SW China region than in Guangxi, with a larger improvement of the RMSE owing to the smaller leakage error at the larger spatial scale of SW China.

The multiplicative method is very sensitive to the a priori model simulations. For all the five a priori model simulations, the multiplicative method results in a reduced correlation coefficient and increased RMSE between leakage-corrected TWS and in situ observations, indicating that the multiplicative correction method incurs errors and is not applicable to leakage correction of GRACE data over SW China and Guangxi. The reason lies in the fact that the multiplicative correction assumes uniform distribution of mass changes inside a region, and ignores the spatial heterogeneity of the actual mass distribution considering the heterogeneous karst hydrogeology conditions of SW China.

This study assessed the accuracy of three mascon solutions (CSR, JPL and GSFC mascons). Mascon-based TWS is nearly as good as SH-based TWS (with additive and scaling factor correction). This study further demonstrates the accuracy and effectiveness of the mascon solutions for interpreting regional TWS at a scale larger than 200,000 km^2^. The advantage of the mascon solution for large scale applications is revealed considering the exemption of filtering and leakage correction procedures.

## Figures and Tables

**Figure 1 sensors-19-03149-f001:**
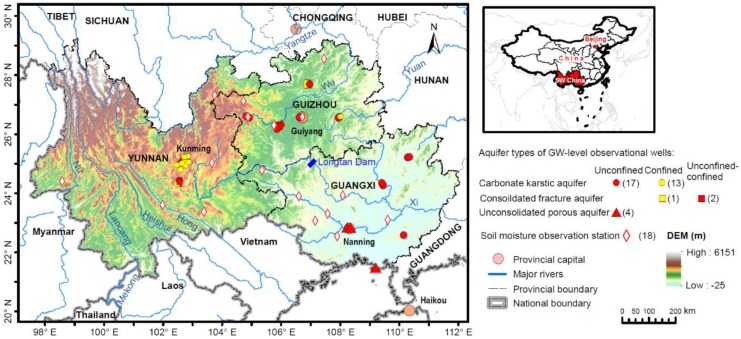
Study region and location of in situ groundwater-level and soil moisture observations.

**Figure 2 sensors-19-03149-f002:**
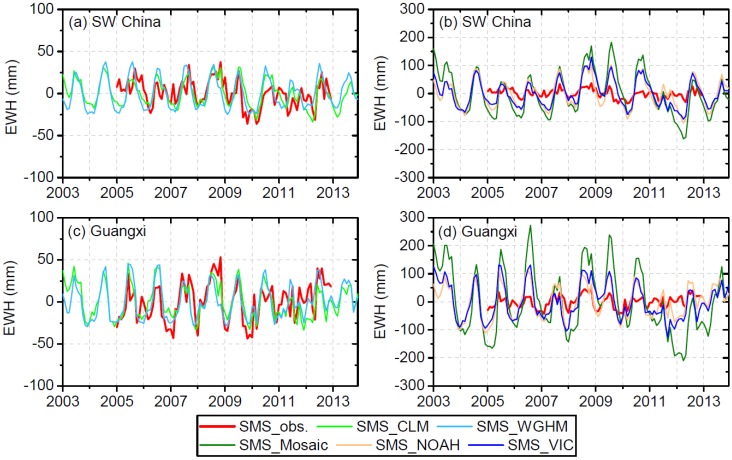
Comparison of simulated SMS with in situ observations of SMS. “Obs.” stands for observed SMS. Water storage anomalies are expressed as equivalent water height (EWH) in mm. Note the difference in the scale of vertical axis.

**Figure 3 sensors-19-03149-f003:**
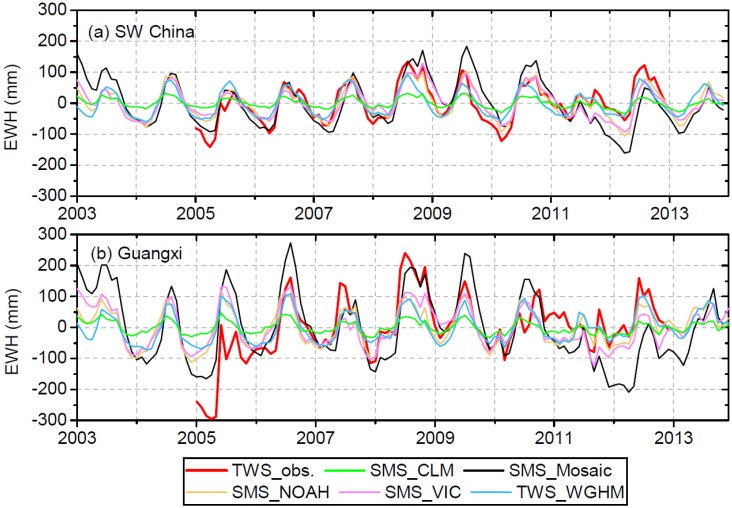
Comparison of simulated SMS from GLDAS models and TWS from WGHM with in situ observations of TWS.

**Figure 4 sensors-19-03149-f004:**
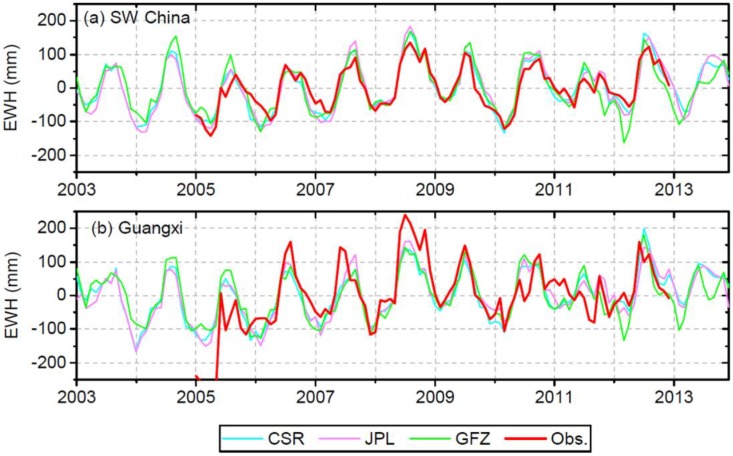
Comparison of uncorrected GRACE TWS from CSR, JPL and GFZ (level-2 SH solutions) with in situ observations of TWS.

**Figure 5 sensors-19-03149-f005:**
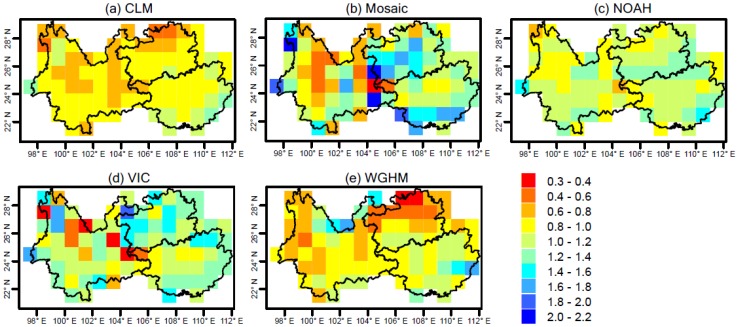
Gridded scaling factors derived using different model simulations as the a priori information. The 0.5° × 0.5° WGHM data are resampled to 1° × 1° as GLDAS models.

**Figure 6 sensors-19-03149-f006:**
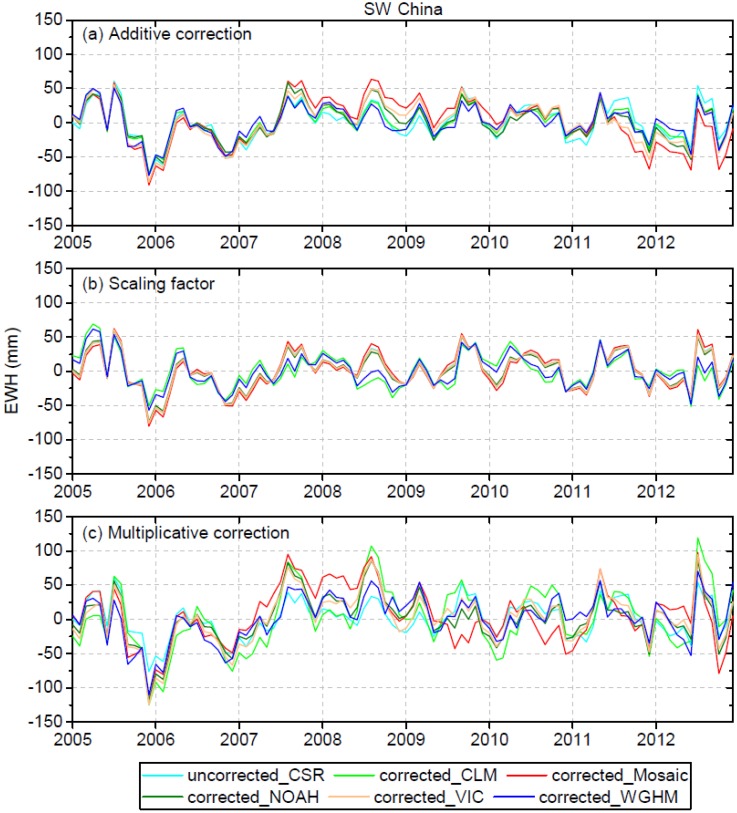
Residuals between uncorrected/leakage-corrected GRACE TWS and in situ observations of TWS for SW China. The leakage-corrected TWS is based on three different methods using the five model simulations as the a priori information.

**Figure 7 sensors-19-03149-f007:**
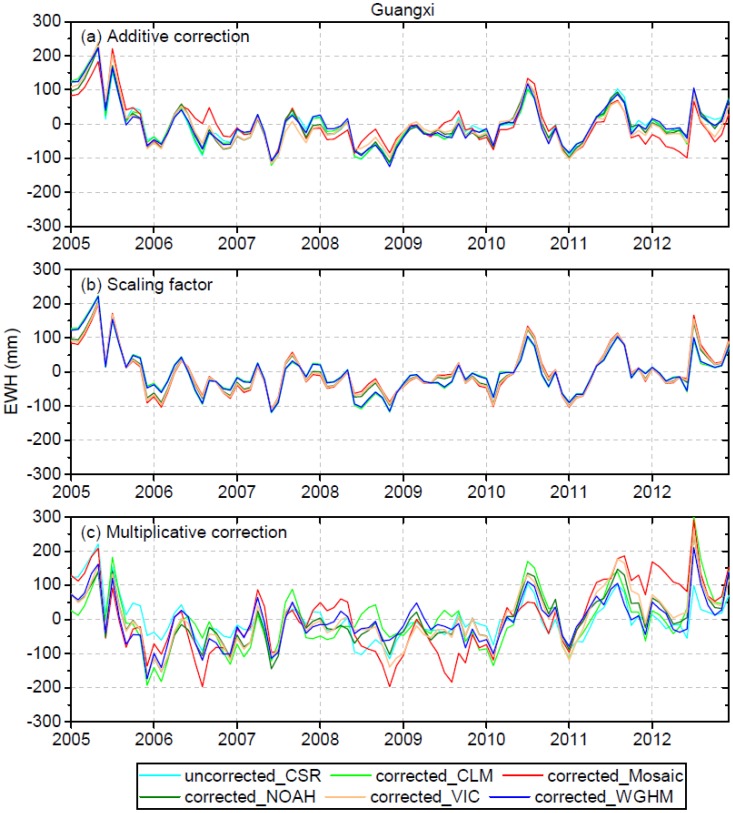
The same as Figure 6, but for Guangxi.

**Figure 8 sensors-19-03149-f008:**
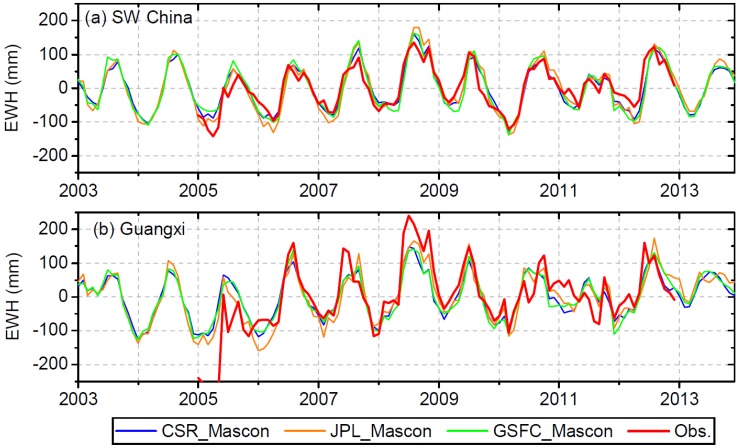
Comparison of mascon-based TWS with in situ observations.

**Table 1 sensors-19-03149-t001:** Statistics of r and root mean square error (RMSE) between in situ SMS and modelled SMS, between in situ TWS and modelled SMS, and between in situ TWS and GRACE-derived TWS. All the correlations are evaluated using a two-sided 1% level significance (i.e., *p* < 0.01).

**Water Storage**	**Model**	**In Situ SMS in SW China**		**In Situ SMS in Guangxi**	
		**r**	**RMSE (mm)**	**r**	**RMSE (mm)**
SMS	CLM	0.63	13.5	0.64	18.5
SMS	Mosaic	0.43	71.0	0.38	108.2
SMS	Noah	0.59	44.1	0.70	45.4
SMS	VIC	0.58	42.2	0.55	54.3
SMS	WGHM	0.46	17.6	0.61	19.4
**Water Storage**	**Model/GRACE**	**In Situ TWS in SW China**		**In Situ TWS in Guangxi**	
		**r**	**RMSE (mm)**	**r**	**RMSE (mm)**
SMS	CLM	0.84	50.3	0.59	91.1
SMS	Mosaic	0.71	54.4	0.61	97.0
SMS	Noah	0.87	30.8	0.66	76.7
SMS	VIC	0.79	38.9	0.59	82.0
TWS	WGHM	0.86	34.8	0.64	79.1
TWS_uncorrected	GRACE CSR	0.93	27.0	0.79	63.1
TWS_uncorrected	GRACE JPL	0.93	29.7	0.80	61.7
TWS_uncorrected	GRACE GFZ	0.86	37.2	0.71	71.1

**Table 2 sensors-19-03149-t002:** Regionally integrated and spatially averaged scaling factors in SW China and Guangxi. Relative difference is defined as the absolute relative difference in scaling factor between the regionally integrated and spatially averaged approaches with respect to the scaling factor for the regionally integrated approach.

Region	Scaling Factor	CLM	Mosaic	Noah	VIC	WGHM	CV
SW China	regionally integrated	0.72	1.04	0.97	0.99	0.79	0.15
SW China	spatially averaged	0.88	1.13	1.09	1.14	0.92	0.12
	Relative difference	22%	9%	12%	15%	16%	-
Guangxi	regionally integrated	0.97	1.34	1.23	1.28	1.01	0.14
Guangxi	spatially averaged	0.99	1.21	1.12	1.17	1.06	0.08
	Relative difference	2%	10%	9%	9%	5%	-

**Table 3 sensors-19-03149-t003:** Statistics of r, RMSE and Nash–Sutcliffe efficiency coefficient (NSE) between leakage-corrected and in situ TWS, and between mascon-based and in situ TWS. All the correlations (r) are evaluated using a two-sided 1% level significance (i.e., *p* < 0.01).

TWS Anomalies	Model/GRACE	SW China			Guangxi		
		r	RMSE (mm)	NSE	r	RMSE (mm)	NSE
Uncorrected	CSR	0.93	27.0	0.82	0.79	63.1	0.61
Additive	CLM	0.93	25.0	0.84	0.79	63.0	0.61
	Mosaic	0.89	34.2	0.70	0.80	60.9	0.64
	Noah	0.93	26.7	0.82	0.78	63.1	0.61
	VIC	0.91	29.4	0.78	0.78	64.1	0.60
	WGHM	0.93	24.3	0.85	0.79	62.6	0.62
Scaling Factor	CLM	0.93	24.8	0.84	0.79	63.5	0.61
	Mosaic	0.93	28.5	0.79	0.79	64.4	0.60
	Noah	0.93	26.0	0.83	0.79	63.0	0.61
	VIC	0.93	26.8	0.82	0.79	63.5	0.61
	WGHM	0.93	23.8	0.86	0.79	63.1	0.61
Multiplicative factor	CLM	0.92	44.9	0.49	0.78	82.6	0.34
	Mosaic	0.84	41.5	0.56	0.54	100.3	0.02
	Noah	0.91	35.4	0.68	0.76	73.4	0.48
	VIC	0.90	37.2	0.65	0.73	79.2	0.39
	WGHM	0.90	32.9	0.73	0.79	66.9	0.57
Mascon	CSR	0.92	25.8	0.83	0.78	64.9	0.59
	JPL	0.92	31.5	0.75	0.82	58.3	0.67
	GSFC	0.89	32.9	0.73	0.77	65.4	0.58

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
