# Peer review of "Sensitivity Analysis of Leakage Correction of GRACE Data in Southwest China Using A-Priori Model Simulations: Inter-Comparison of Spherical Harmonics, Mass Concentration and In Situ Observations"

_sensors, 2019, doi:10.3390/s19143149_

Round 1
Reviewer 1 Report
The study is well presented but language errors are present. A thorough review addressing the grammatical mistakes and better sentence formulation is recommended. There are a few minor comments/suggestion:
Line 80-81: A recent study demonstrated that the quality of mascon solution is better at catchment scale but the spherical harmonic solution, when processed efficiently, can perform better. Citing this paper and using its results to motivate readers is recommended. Lan Zhang, Shuang Yi, Qiuyu Wang, Le Chang, He Tang, Wenke Sun, Evaluation of GRACE mascon solutions for small spatial scales and localized mass sources, Geophysical Journal International, Volume 218, Issue 2, August 2019, Pages 1307–1321, https://doi.org/10.1093/gji/ggz198
Table 3 contains correlation coefficients and RMSE, I would urge authors to add NSE (Nash Sutcliffe efficiency) to this table. Correlation alone or with RMSE can not help us quantify the quality of results. NSE is one metric that takes correlation, bias, and error into account. (https://en.wikipedia.org/wiki/Nash%E2%80%93Sutcliffe_model_efficiency_coefficient). Hopefully this will help authors to getter a comprehensive idea about quality of different GRACE outputs.
It is a bit hard to agree that the Mascons have done a better job overall. From Table 3, one can say that for SW china mascons are nearly as good as other SH based solutions. Please revisit the table and discussion carefully.
Author Response
Reviewer #1
The study is well presented but language errors are present. A thorough review addressing the grammatical mistakes and better sentence formulation is recommended. There are a few minor comments/suggestion:
Line 80-81: A recent study demonstrated that the quality of mascon solution is better at catchment scale but the spherical harmonic solution, when processed efficiently, can perform better. Citing this paper and using its results to motivate readers is recommended. Lan Zhang, Shuang Yi, Qiuyu Wang, Le Chang, He Tang, Wenke Sun, Evaluation of GRACE mascon solutions for small spatial scales and localized mass sources, Geophysical Journal International, Volume 218, Issue 2, August 2019, Pages 1307–1321, https://doi.org/10.1093/gji/ggz198
Response: Revised as suggested. See Line 82-85.
Table 3 contains correlation coefficients and RMSE, I would urge authors to add NSE (Nash Sutcliffe efficiency) to this table. Correlation alone or with RMSE cannot help us quantify the quality of results. NSE is one metric that takes correlation, bias, and error into account. (https://en.wikipedia.org/wiki/Nash%E2%80%93Sutcliffe_model_efficiency_coefficient). Hopefully this will help authors to getter a comprehensive idea about quality of different GRACE outputs.
Response: Revised as suggested. See Line 386 and 399-404.
It is a bit hard to agree that the Mascons have done a better job overall. From Table 3, one can say that for SW china mascons are nearly as good as other SH based solutions. Please revisit the table and discussion carefully.
Response: We further revised the abstract (see Line 42-43) as well as conclusion (see Line 502-506).
Reviewer 2 Report
Dear authors,
This paper present an interesting methodologie based on the sensitivity analysis (performed with observational data in the Southwest China and one sub-region Guangxi) of three leakage correction methods of GRACE data, normally obtained with forward modelling based on some a-priori information provided by different global hydrological models, that could be very helpful to choose the most appropriate method for leakage correction in other study area.
Author Response
Reviewer #2
Dear authors,
This paper present an interesting methodologie based on the sensitivity analysis (performed with observational data in the Southwest China and one sub-region Guangxi) of three leakage correction methods of GRACE data, normally obtained with forward modelling based on some a-priori information provided by different global hydrological models, that could be very helpful to choose the most appropriate method for leakage correction in other study area.
Response: We thank the reviewer for the evaluation of our manuscript.
Reviewer 3 Report
General comments:
This study conducted the sensitivity analysis of three leakage correction methods to five global hydrological model simulations in the Southwest China (at a typical karst region with complex hydrogeology conditions) by comparing the performance of five global hydrological models, three leakage correction strategies and the mascon solutions. This study contributes to understanding the sensitivity of leakage correction of GRACE level-2 data to various a-priori model simulations as well as the performance of leakage corrected TWS based on the level-2 SH solution in comparison with the level-3 mascon. In general, this manuscript is interesting to me and can be accepted with minor revisions.
Minor comments:
1. A proper description of the forward modelling is suggested. In this study, the description of the forward modelling and its advantages is inadequate.
2. It is unclear why this study additionally chooses the sub-region Guangxi, maybe proper explanation should be provided.
3. The main source of uncertainties should be discussed in this study, such as the in situ observations.
4. The meaning of comparing two types of scaling factors, spatially averaged and regionally-integrated, is unclear. An adequate explanation is needed.
5. TWS means total terrestrial water storage anomaly while SMS means soil moisture storage in this paper. For comparison, it is suggested to change the meaning of SMS to soil moisture storage anomaly.
Author Response
Reviewer #3
General comments:
This study conducted the sensitivity analysis of three leakage correction methods to five global hydrological model simulations in the Southwest China (at a typical karst region with complex hydrogeology conditions) by comparing the performance of five global hydrological models, three leakage correction strategies and the mascon solutions. This study contributes to understanding the sensitivity of leakage correction of GRACE level-2 data to various a-priori model simulations as well as the performance of leakage corrected TWS based on the level-2 SH solution in comparison with the level-3 mascon. In general, this manuscript is interesting to me and can be accepted with minor revisions.
Response: We thank the reviewer for the evaluation of our manuscript.
Minor comments:
1. A proper description of the forward modelling is suggested. In this study, the description of the forward modelling and its advantages is inadequate.
Response: The description of forward modelling is presented at Line 72-74. One more sentence is added to address the advantage of forward modelling (see Line 74-76).
2. It is unclear why this study additionally chooses the sub-region Guangxi, maybe proper explanation should be provided.
Response: Revised. See Line 133-138.
3. The main source of uncertainties should be discussed in this study, such as the in situ observations.
Response: The uncertainties for estimating in situ total water storage anomalies are estimated. Please see Section 2.4, Line 202-209.
4. The meaning of comparing two types of scaling factors, spatially averaged and regionally-integrated, is unclear. An adequate explanation is needed.
Response: Explanation is added at Line 366-367.
5. TWS means total terrestrial water storage anomaly while SMS means soil moisture storage in this paper. For comparison, it is suggested to change the meaning of SMS to soil moisture storage anomaly.
Response: The abbreviation of SMS is revised at Line 161.
This manuscript is a resubmission of an earlier submission. The following is a list of the peer review reports and author responses from that submission.